# Beyond the perception of present economic inequality: How people construe past, present and future wealth gaps

Andrea Scatolon[1]*, Silvia Galdi[2], Carmen Cervone[1], Lucia Mannetti[3], Anne Maass[4]

1 Department of Developmental Psychology and Socialisation, University of Padua, Italy, 2 Department of Psychology, University of Campania "Luigi Vanvitelli", Italy, 3 Department of Psychology of Developmental and Socialization Processes, Sapienza University of Rome, Italy, 4 Department of Psychology, New York University Abu Dhabi, United Arab Emirates

* andrea.scatolon@unipd.it

## Abstract

People tend to misperceive economic inequality levels in their country. However, studies investigating such misperceptions focused mainly on appraisals of present inequality. In two cross-sectional studies ($N_{total} = 600$), we investigated perceived past, present, and future wealth inequality in Italy, as well as accuracy in ranking EU countries by perceived wealth inequality. Confirming past literature, participants underestimated present levels of national wealth inequality. Results also showed the existence of a decline model in estimations of wealth inequality (i.e., optimism towards past, pessimism towards future), such that participants consistently expected an increase of future inequality (roughly corresponding to actual trends). Interestingly, participants were relatively accurate in ordering European countries, including Italy, with respect to levels of wealth inequality. We interpret these findings as reflecting a rational mechanism, where participants linearly project how wealth inequality grows (or declines) over time, and then plot their estimates for specific points in time on such linear projection. We therefore suggest that such schema prompt relative (rather than absolute) evaluations of wealth inequality, which leads to increased accuracy. We conclude that future interventions should accommodate people's mental reconstruction of inequality.

## Introduction

Wealth inequality is on the rise worldwide, from the most industrialized and individualistic nations such as the US, to ex-communist and collectivist nations such as China [1]. This trend towards increasing economic inequality has been observed since the 80ies and has recently been amplified by the COVID-19 pandemic [2]. Compared to the past, wealth inequality in OECD countries has reached "its highest level since the past 40 years" [3], and "more than 70 per cent of the world population now live[s]

**Data availability statement:** All files (data, questionnaires and supporting information) are available from the OSF database (https://osf.io/p69re/?view_only=ed44a73a32d64aa886c-4764c8484d4d1).

**Funding:** This research was supported by grant PRIN 2017 #2017924L2B of the Italian Ministry of Education, University and Research (MIUR - https://www.mim.gov.it/), entitled "The psychology of economic inequality", to the last author (AM). The funders did not play any role in the study design, data collection and analysis, decision to publish, or preparation of the manuscript.

**Competing interests:** The authors have declared that no competing interests exist.

in countries where income inequality has increased in the last three decades" [4]. Looking forward to the future, income inequality in 2050 could either further escalate or slightly decline depending on whether nations were to follow the US or European countries, respectively [1].

Given the relevance of this issue, one would logically assume that individuals should have some understanding of wealth distribution in their country. Yet, many studies have highlighted that laypeople tend to misperceive current levels of wealth inequality [5–7]. While informative, however, research on inequality (mis)perception has specifically focused on *present* levels of wealth inequality, rarely comparing current levels of inequality to either past situations or potential future scenarios. Overall, these studies have effectively left an interesting question unexplored: how do individuals perceive time trajectories of wealth inequality? Do people have a general grasp of changes in wealth inequality in their country?

The present research aimed at both confirming past literature on (mis)perception of wealth inequality in the specific context of Italy (where the top-20% of the population owns about 70% of national wealth; [8]) while also expanding on it by investigating perceived past and future levels of wealth inequality. For the latter, we drew inspiration from literature on collective memory biases [9–11], and propose the existence of an optimistic bias in relation to past wealth inequality, and an opposite, pessimistic bias when envisioning future wealth distributions [12].

### Past and future collective memory biases

Research on personal temporal thought shows that individuals display a generalized positivity bias, both when remembering past events and when imagining future scenarios in which they are personally involved. This bias presumably serves as a coping function for maintaining one's well-being and self-identity [13,14]. A more recent line of research on collective temporal thought [10] has distinguished between two interconnected processes: the personal and collective temporal thought. The collective temporal thought includes individuals' recollections of past events (i.e., collective memory [14]); and their anticipation of future events (i.e., collective future thought [15,16]); that are relevant to, and shared by, the group they belong to.

Personal and collective temporal thoughts work differently: even though people show a positive bias in recollecting personal memories and imagining personal future scenarios, they are more pessimistic about collective past and future events [12,17]. Consistently, recent research on perception of risks related to economic issues [12] has revealed that participants showed a pessimistic bias when future risks were framed as involving the fate of their country (e.g., number of families with economic problems), while the opposite was true when considering corresponding personal risks (e.g., personally facing economic problems). Such findings suggest that a similar pattern could emerge when assessing beliefs about future disparities in wealth.

While collective temporal thought can be studied in relation to group-shared events of any sort, interest has mainly revolved around country narratives. In this regard, research has evidenced two possible patterns when picturing collective events across time. Some studies documented a decline pattern, showing that US and

French citizens manifested a positive bias in recollecting national events relevant to the past (consistent with [18,19]), and had negative perspectives for the future (consistent with [17]) of their country [10,11]. Other studies, instead, have found an improvement pattern, such that US citizens were pessimistic about the national collective past but slightly optimistic when imagining the national collective future [9].

Similar trajectories towards improvement or decline are well documented in autobiographical memories as well [20–22]. According to this literature, the past is reconstructed on the basis of implicit lay theories about improvement versus decline. The (known) present state of affairs provides the anchor for reconstructing the past, whereas lay theories determine whether the reconstruction of the past (and projection into the future) will be positive or negative. For instance, people may overestimate their past achievements if they have a theory of decline, but they may underestimate their past achievements when believing that their abilities have improved over time. Drawing from this evidence, one may envisage a similar process in the context of wealth inequality, such that people will use the (perceived) current state of inequality as an anchor and adjust the past and future estimates according to their belief that inequality is increasing or decreasing.

Previous studies on collective memory biases have mainly compared past and future events but failed to consider current times as an additional step to assess changes across time. When considering national events or issues (wealth inequality, in our case), however, the present may facilitate time comparisons. Furthermore, past research on collective memory biases focused on non-specific events, usually asking participant to spontaneously recall (for the past) or imagine (for the future) any event related to their country (e.g., [9,10]). Among others, wealth inequality is a widely discussed issue, extensively debated by scholars [1,23], media [24], and laypeople (as for instance illustrated by Google Trends' data showing an increasing search for the keyword "economic inequality"; see Supporting Information for a graphical representation S1 File) alike. We therefore propose that expanding research on memory biases to this collectively relevant topic would represent a novel and effective way to assess the existence and direction of these biases.

### Estimation of wealth inequality

Research conducted in the US showed how individuals systematically underestimate actual inequality levels in their own country, even though they are aware of their increase with time [25]. Additional research highlighted that people underestimate existing economic inequality in their country and that they desire a much more equal society ( [7]; but see [26], for an exception). This result pattern was replicated not only in the US among adult [27] and adolescent samples [28], but also in different countries, such as Australia [29]. A similar pattern was found for income inequality that is underestimated in most countries and that is consistently considered as too large [30].

Although it is well established that people are aware of the existence of some wealth inequality in their country, and that individuals' perceptions are modeled by several factors, such as system-justifying beliefs [31], little is known about their awareness of either past or future inequality. Chambers and colleagues [26] partly addressed this issue for past income (rather than wealth) inequality. Participants were provided with information on the ratio between top and bottom income quintiles in 1967 and asked about top versus bottom income distributions, both current and across several past decades (from 1970 to 2010). They found that participants overestimated how income inequality rose over time. It should be noted that, since the authors provided an anchor when assessing participants' perceptions of inequality, this design does not allow for a measurement of potential *lay* memory biases – which, as we argue in the section detailing the aims of our research, represent a more ecologically sound strategy to conduct research on the topic.

Indirect evidence of future wealth inequality awareness comes from recent research on perception of risks related to economic issues [12]. Participants showed a pessimistic bias when future risks were framed as involving the fate of their country (e.g., number of families with economic problems), while the opposite was true when considering corresponding personal risks (e.g., successfully managing economic problems). These findings are in line with the previously mentioned disconnection between personal and collective temporal thought [32], thus suggesting that a similar pattern could emerge when assessing beliefs about future disparities in wealth.

Together, these studies suggest that laypeople are rather poor at judging existing levels of inequality, although they seem to be aware of the increase of inequality over time [26]. Thus, although their estimates are greatly inaccurate in absolute terms, their relative judgments (time trajectory) are actually much closer to reality. However, perceived past, present, and future inequality were never investigated within the same study: therefore, previous research does not have the ability to accurately assess participants' projections of inequality over time. Furthermore, assessing simultaneously past, present, and future inequality estimations also provides us with insights on the potential psychological mechanisms underpinning these patterns. If, indeed, participants believed wealth inequality to be simultaneously lower in the past and higher in the future, this would point to a collective memory bias being at play. To our knowledge, this is the first research bridging collective memory biases and perception of wealth inequality. Based on previous literature, we have reasons to believe that such a connection exists: for example, consider how previous literature [11] has provided evidence for a decline model (which we intend as referring to an increase in inequality as a negative societal trend) when assessing memory biases, and how a specific variable, namely perceived societal anomie (i.e., perceived deregulation and disintegration of one's society), worked as a buffer for such biases. As this variable is positively associated with perceived economic inequality [33], it is logical to assume that a similar pattern of decline could also be found when assessing perceived past, present, and future inequality.

## The present research

The main aim of the present work was to investigate perceptions of wealth inequality across time, by asking Italian participants to estimate past, present, and future wealth inequality, in order to evaluate positive and negative collective biases towards past and future, respectively [10].

## Aims and hypotheses

We suspect that people may be able to understand inequality in relative terms, even where their absolute judgments are inaccurate. Relative judgments tend to be more accurate than absolute ones [34], and this general principle holds also for socio-economic questions such as the perceptions of social mobility [35]. Extending this reasoning to the perception of economic inequality, we here propose that lay people may be unable to grasp the actual magnitude of wealth gaps, yet they may show a reasonable intuition for whether wealth inequality is declining or increasing. Therefore, our research investigates such relative judgements in addition to absolute ones.

There are at least three reasons to hypothesize that participants will endorse a "decline theory", according to which wealth inequality is getting worse. First, the Italian Gini index has increased steadily since 2007 [36,37] and people may be aware of this increase in inequality. Second, wealth inequality represents a collective problem that occurs outside of people's personal control. The lack of personal control is known to produce more pessimistic outlooks and explains why people are generally optimistic about their personal future but pessimistic about their collective future [9,12]. Third, Italians tend to be more pessimistic about the present and future national economy, compared to most other European citizens [38,39].

Note that we use the term "decline" to refer to an increase in perceived wealth inequality. While not everyone may agree with the fact that inequality is a negative occurrence (see [40], here we refer to a "decline" in line with literature emphasizing the relationship between inequality and numerous health, social, and environmental issues [41].

Therefore, drawing from the reviewed literature, we hereby outline 2 main hypotheses, which were tested with an exploratory approach in Study 1, and a confirmatory approach in Study 2 (preregistered at https://osf.io/p69re/?view_only=ed44a73a32d64aa886c4764c8484d4d1).

*Hypothesis 1*. Participants would underestimate actual levels of wealth inequality (*Hypothesis 1a*); participants' ideal wealth distribution would be more equal than both the actual and estimated wealth distribution (*Hypothesis 1b*), but participants' ideal distribution would exceed an equal distribution (20% for each quintile of the population – based on Norton and Ariely's "equal country" scenario; (7)); *Hypothesis 1c*).

Moreover, following the decline model [10], we hypothesized that

*Hypothesis 2*. Participants would correctly perceive an increase in wealth inequality over time, estimating past wealth inequality to be smaller than present inequality (*Hypothesis 2a*); participants would also be pessimistic about the future development, predicting an increase in wealth inequality in the future (*Hypothesis 2b*).

Data, questionnaires and Supporting Information (S1 File) including more detailed information on measures are available on OSF: https://osf.io/p69re/?view_only=ed44a73a32d64aa886c4764c8484d4d1.

## Study 1

### Method

**Participants.** The procedure and materials of the study were approved by the Ethics Committee for Psychological Research of the University of Padua (protocol #1860). Data collection was run between 24th March 2016 and 26th September 2016. Participants were contacted through social networks (i.e., Facebook) and snowball sampling (including fellow students, neighbors, acquaintances, etc.; see [12], for a similar procedure) and asked to take part in a study aimed at investigating individuals' attitudes and beliefs about wealth inequality in Italy. Those who agreed to participate in the study were linked to the study URL where they provided a written informed consent and then started the survey. The final sample consisted of 354 participants, 90 (25%) residents of Northern (38 men; 52 women), 127 (36%) residents of Central (62 men; 65 women) and 137 (39%) residents of Southern Italy (62 men; 75 women). All participants were Italian, and their age ranged from 18 to 78 years ($M = 30.12$, $SD = 10.59$). Most respondents had a high school ($n = 137$; 39%) or a university degree ($n = 179$; 56%), whereas 9 participants (3%) had the lowest formal qualification. Two hundred fourteen participants (66%) self-defined as liberals. The results of a post-hoc sensitivity power analysis ran on G*Power 3 [42] showed that our sample had 80% statistical power to detect an effect size $f \geq .07$ for a repeated measures ANOVA, within factors (3 measurements).

**Procedure and measures.** The questionnaire was developed through SurveyMonkey (https://www.surveymonkey.com). After consenting to participate in the study, participants rated how they felt about their lives at the present time (i.e., life satisfaction; see supplemental material (S1 File)) and provided information about gender, age, profession, level of education, monthly income, and political orientation. Then, participants performed three tasks aimed at assessing their perception of wealth distribution in Italy in the present, in the past, and in the future. Respondents then completed the second part of the questionnaire, which evaluated: (a) wealth distribution in different European countries; (b) beliefs about causes of wealth inequality in Italy; (c) personal economic hardships and sources of information mainly used to gather knowledge on socio-economic inequalities (see supplemental material (S1 File)). At the end of the survey, participants were fully debriefed and thanked for their participation. Responding to the questionnaire required approximately 10 minutes.

**Demographics**. Since life satisfaction is negatively affected by levels of wealth inequality of a country (e.g., [43]), two items were used to measure life satisfaction as a control variable (i.e., satisfaction with life in general and satisfaction with the economic situation, both assessed on a Likert scale from 1 = *not at all satisfied* to 5 = *completely satisfied*). Participants answered demographic questions including gender, age, education, occupational status, monthly household net income, number of household members, region of origin and of residence, sexual orientation, political orientation, and religiosity.

**Wealth inequality estimation task.** A modified version of the Norton and Ariely's measure [7] was used to investigate individuals' perceptions of the distribution of wealth in their country. The present version allows participants to focus only on the top and bottom quintile of the population and provides a multiple-choice response instead of a self-generated distribution of wealth. Such decision was deemed necessary in order to both avoid difficulties arising from the original method, in line with criticisms of some scholars (e.g., [6,44]), to reduce memory biases, anchoring biases, and cognitive load on respondents, and to make the task more intuitive and overall easier to understand (for additional details, see "A Methodological Note" on the Supporting Information S1 File).

To help participants with the task, we ensured that all had the same working definition of economic quintiles. Before beginning the task, respondents were shown a picture portraying five rows, each including 20 people, and read the following: "The following picture represents the totality (100%) of the Italian population, divided into 5 social classes. Each row represents a specific social class, from the richest to the poorest one. To answer this questionnaire, you will need to refer to two of these social classes: A) the 'rich', namely the line of individuals colored in BLUE, which represents the top 20% of the population; B) the 'poor', namely the line of individuals colored in RED, which represents the bottom 20% of the population".

Immediately afterwards, respondents performed the task. As can be seen in Fig 1, we created nine pie charts representing nine different wealth distributions: from a perfectly equal distribution to a highly unequal one. Each pie chart showed the percentage of wealth held by the two most extreme economic quintiles of population, with the top 20% colored in blue and the bottom 20% colored in red, as well as the total percentage of wealth held by the remaining second, third, and fourth quintile colored in white. Unbeknownst to respondents, one of the nine charts reflected the actual wealth distribution in Italy in 2015 (i.e., Fig 1, chart C; Oxfam, 2016). For technical matters, we were able to develop only one pie chart with a clearly representable wealth distribution that was even more unequal than the current one (i.e., Fig 1's chart

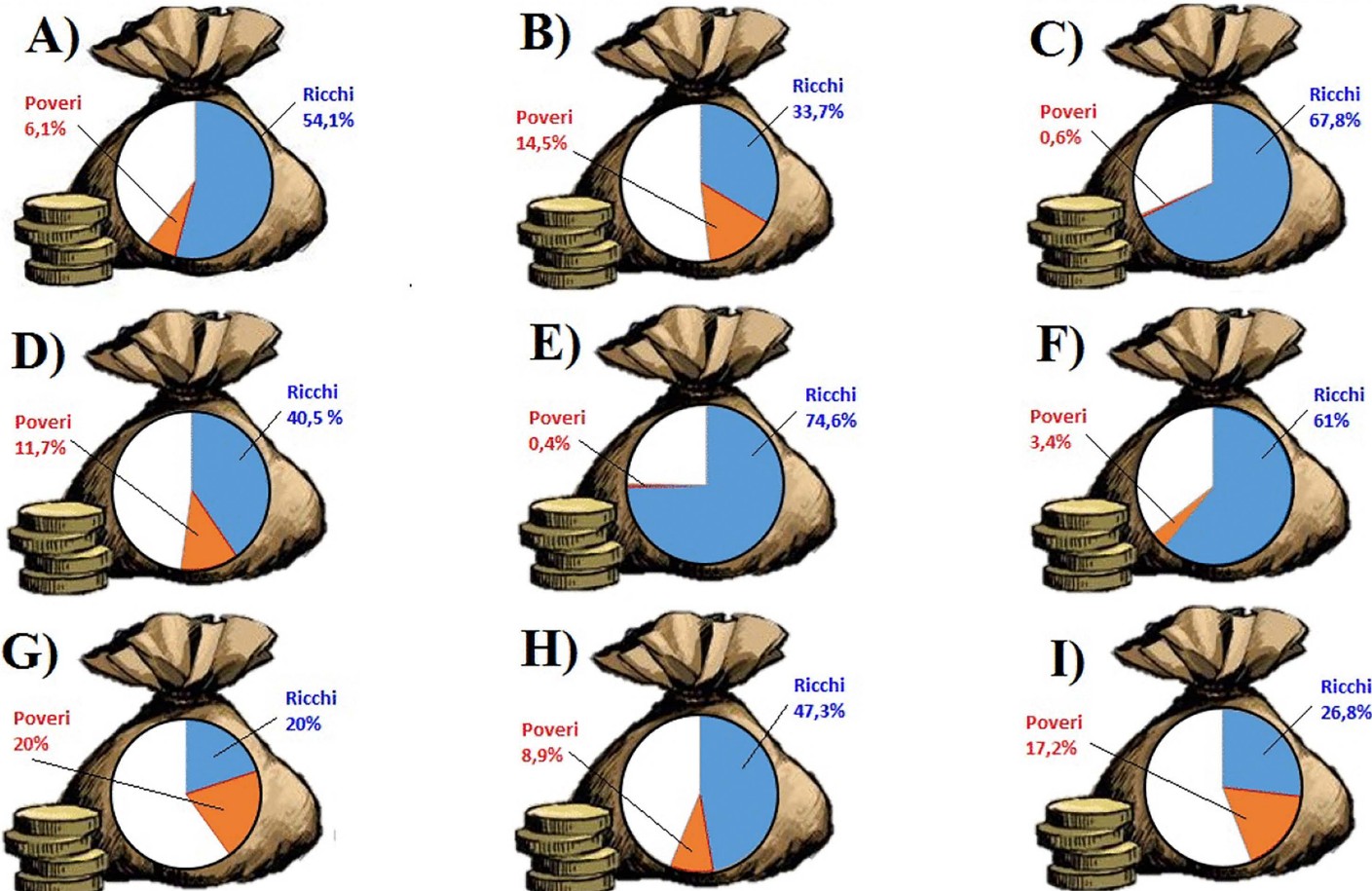

**Fig 1. Wealth inequality estimation task – stimuli (Study 1 and Study 2).** Note: the corresponding graphical representation of the options for estimation of past and future presented identical options in terms of percentages, but with a different order and different colors (i.e., green for top 20%, and yellow for bottom 20%).

E), whereas the remaining seven charts portrayed ever more equal distributions (e.g., Fig 1's chart A). Therefore, participants were presented with the nine pie charts in random order and read the following: "Each graph represents the totality of Italy's wealth and shows the percentage of wealth held by the wealthiest (colored in blue) and the poorest (colored in red) quintile of Italian population. The richest quintile owns the blue slice of the chart, while the poorest one owns the red slice of the chart". The set of pie charts was repeated four consecutive times, asking participants to indicate which of the nine graphs pictured respectively: (a) Italy's *present* wealth distribution (estimation of present inequality); (b) the *ideal* distribution of wealth (ideal inequality); (c) Italy's wealth distribution in the *past* (i.e., "ten years ago"; estimation of past inequality); (d) Italy's wealth distribution in the *future* (i.e., "ten years from now"; estimation of future inequality). For past and future wealth inequality, a tenth textual choice option, indicating that wealth gaps were even higher than the depicted ones, was provided.

**European wealth distribution ranking.** As a secondary aim, we wanted to see whether our argument concerning relative (vs. absolute) accuracy would also hold for a different type of comparative judgment, namely the relative standing of different countries. Therefore, participants were shown a map of Europe and were asked to rank 9 countries (presented in alphabetical order) from the most equal to the least equal in terms of wealth distribution. Nations were selected so as to cover a wide range of wealth inequality levels among European countries. In ascending order of the Gini Index (which, differently from the wealth inequality estimation task, refers to income rather than wealth inequality) in 2015, we selected: Finland (.259), Denmark (.261), Norway (.262), Hungary (.280), Germany (.294), Italy (.327), Greece (.333), Spain (.341) and United Kingdom (.351; see [45]). Given that the main focus is on temporal comparisons, results concerning cross-country comparisons will be presented only in the Supporting Information.

**Causal attributions scale.** Do individuals simply (mis)perceive wealth inequality, or are they influenced by ideological beliefs in doing so? To answer this question, a list of 19 causes of present wealth inequality was developed *ad hoc* by the authors. Causes referred to *internal* (7 items, e.g., "people's scarce economic skills", $\alpha = .82$), *structural* (external; 9 items, e.g., "unequal taxation system", $\alpha = .79$), and *criminality* (external) causes (3 item, i.e., "tax evasion", $\alpha = .73$). Participants indicated how much they deemed each cause to be related to wealth inequality, using a 5-point Likert scale ranging from 1 (*it has no influence at all*) to 5 (*it influences a lot*). Given the primary focus of this paper on the perception of wealth inequality over time, we will not report the findings of this variable in the main text. This variable was adopted purely for exploratory purposes; for details regarding factor analyses of this scale, and for exploratory analyses concerning this variable, see Supporting Information.

## Results

To simplify reporting, we will focus on the estimates of wealth owned by the top quintile of the population, given that the nine pie charts were arranged so that estimates for top and bottom quintiles were interdependent. Adding life satisfaction, personal economic hardship, political orientation, or sources of information as control variables did not affect results and, therefore, we will not discuss these variables further. Participants indicating the "even higher" option ($N = 31$) for past and future wealth inequality were not included in the following analyses, as this response is not quantifiable. Correlational tables are available in the Supporting Information.

### Estimated, ideal, and actual inequality

In support of Hypothesis 1a, participants greatly underestimated the total wealth owned by the top quintile of the population ($M = 46.73\%$, $SD = 17.64\%$ compared to the actual value of 67.8%), one-sample $t(353) = -22.48$, $p < .001$. In support of Hypothesis 1b, participant's ideal distribution ($M = 35.31\%$, $SD = 18.35$) was more equal than their own estimates, paired-sample $t(353) = 7.34$, $p < .001$, $d = .38$. At the same time, in line with Hypothesis 1c, the ideal distribution exceeded a perfectly equal distribution (20%), one-sample $t(353) = 15.70$, $p < .001$, suggesting *t*hat participants desired some degree of wealth inequality.

## Comparing past, present, and future inequality

A repeated measures ANOVA with Greenhouse-Geisser correction showed a significant effect for time (past, present, future), $F(2, 614) = 34.44$, $p < .001$, $\eta^2_p = .10$. The linear trend was significant, $F(1, 322) = 56.01$, $p < .001$, $\eta^2_p = .15$. In line with Hypotheses 2a and 2b, present wealth inequality ($M = 46.47\%$, $SD = 17.37\%$) was estimated as significantly higher than past inequality ($M = 42.47\%$, $SD = 14.10\%$), $t(322) = 3.73$, $p < .001$, $d = .24$. Moreover, present inequality was estimated as significantly lower than future inequality ($M = 51.60\%$, $SD = 18.87\%$), $t(322) = -5.04$, $p < .001$, $d = -.27$.

## Discussion

Results from Study 1 confirmed past findings (e.g., [7,27]), showing that laypeople generally underestimate present levels of wealth inequality in their country. Although respondents perceived the existence of a certain amount of wealth inequality between the top and bottom quintiles of the Italian population, they underestimated its extent and revealed a preference for a society with a more equal, but still somewhat unequal, distribution. Indeed, participants' ideal society corresponded approximately to a ratio of 1 to 2.5, such that the richest quintile of the population would own about two and a half times the wealth of the poorest quintile, quite different from the actual 1–113 ratio.

As for wealth inequality perceptions across time, participants correctly believed wealth inequality to have been lower in the past, and they predicted it to be even higher in the future as compared to the 2015 levels, thus providing first proof for the hypothesized decline model when considering collective national memory biases [10]. Importantly, although wealth distribution estimates were inaccurate in absolute terms, participants were accurate in their understanding of increasing inequality over time. They correctly reported lower inequality in the past and they correctly predicted an increase of inequality in the future. The perceived and predicted time trend indicated by our participants seems to mirror the World-bank Gini Index, which did indeed increase between 2006 (Gini = 33.7) and 2016 (Gini = 35.2) and is predicted to show a further increase by 2026 (predicted Gini = 36; [46]).

It should be pointed out that participants may have believed present wealth inequality levels to be much higher but could not express so due to measurement constraints (i.e., participants had only one option that was more unequal than present wealth inequality levels). Indeed, when they had more possibility to express greater pessimism (i.e., the European ranking task), they did so, by ranking Italy as more unequal than it actually was compared to other countries.

## Study 2

To test the robustness of the above results, we replicated the study in more recent times (namely 2021), which are also characterized by different issues than those experienced in 2016, such as the COVID-19 pandemic. Study 2 was run five years after Study 1. Importantly, this not only allowed us to evaluate whether the estimate of past and present inequality was accurate, but also whether or not it coincided with the prior estimates of participants Study 1.

Study 2 was almost identical to Study 1. We tested whether participants would desire an ideal wealth distribution that is more equal than their estimated present wealth distribution (Hypothesis 1a), but that would exceed an equal distribution (20% for each quintile; Hypothesis 1b). As the response options were identical in the two studies, underestimation of actual levels of wealth inequality compared to actual (i.e., 2021) data could not have been tested, as the actual level of inequality in 2021 was not among the response alternatives. We further tested the decline model, predicting that participants would estimate past wealth inequality to be smaller than present wealth inequality (Hypothesis 2a), while future wealth inequality would be higher (Hypothesis 2b). The present study was preregistered (https://aspredicted.org/V72_YKN).

### Method

**Participants.** The procedure and materials of the study were approved by the Ethics Committee for Psychological Research of the University of Padua (protocol #1860). Data collection was run between 22nd April 2021 and 23rd April

2021. Participants were recruited through Prolific Academic and payed 1£ for their participation in the study. They provided a written informed consent and then started the survey. As stated in our preregistration, we determined sample size through G*Power 3 [42], which indicated 266 participants to detect an effect size of $R^2 = .04$ (i.e., the variance explained by the causal attributions on ranking of Italy, see Supporting Information for additional details) with 80% power and $\alpha = .05$, in a regression model with three predictors. Two hundred and ninety participants opened the online survey; after exclusion according to our preregistered criteria (i.e., participants who did not complete the questionnaire, those who did not give consent to data processing, and those who failed at least one attention check; $N_{total} = 44$), the final sample consisted of 246 participants, 111 (45%) residents of Northern (68 men; 42 women), 40 (16%) residents of Central (28 men; 12 women) and 95 (39%) residents of Southern Italy (69 men; 22 women, 3 non-binary people). Participants' age ranged from 18 to 63 years ($M = 27.73$, $SD = 9.31$). Most respondents had a high school ($n = 148$; 60%) or a university degree ($n = 87$; 36%), whereas 6 participants (2%) had the lowest formal qualification. One hundred and eighty-two participants (74%) self-defined as liberals.

### Procedure and measures

Study 2 procedure followed the one presented for Study 1, with only a few differences aimed at shortening the question-naire. Items investigating participants' sexual orientation, life satisfaction, personal economic hardships, and sources of information were removed because they did not affect results in Study 1. More importantly, estimated wealth distribution was assessed by using the same stimuli as Study 1 but changing the time interval from 10 to 5 years for the estimation of past and future wealth inequality. Since we were interested in comparing participants' estimates across our 2016 and 2021 data collections, in the present Study 2 the estimation of past wealth inequality referred to 2016 (i.e., "5 years ago"), whereas the estimation of future wealth inequality referred to the same moment in time of Study 1 (2026; i.e., "5 years from now"). The European wealth distribution task was identical to Study 1; in ascending order of the Gini Index in 2021, the countries are ordered as follows: Denmark (.268), Finland (.273), Hungary (.278), Norway (.285), Germany (.303), Greece (.312), Spain (.320), Italy (.330), and United Kingdom (.354; see [45]). Please note that, due to lack of data for 2021, Germany and Denmark Gini indexes were taken from 2020 data. As for Study 1, results concerning cross-country comparisons and causal attributions will be presented only in the Supporting Information.

### Results

As in Study 1, we focused on the estimates of wealth owned by the top quintile of the population and excluded participants ($n = 13$) indicating the "even higher" option for past and future wealth inequality. As preregistered, all analyses were run on the full sample, after excluding one participant who had a completion time shorter than one third of the median. Since no significant differences emerged between the two samples, below we report analyses on the full sample (see Supporting Information for details). Correlational tables are available in the Supporting Information.

### Estimated and ideal inequality

In support of Hypothesis 1a, participant's ideal levels ($M = 30.77\%$, $SD = 14.95\%$) were lower than their estimated present levels of wealth inequality ($M = 49.11\%$, $SD = 15.99\%$), paired-sample $t(245) = 12.25$, $p < .001$, $d = .81$. At the same time, in line with Hypothesis 1b, the ideal wealth distribution exceeded a perfectly equal distribution (20%), one-sample $t(245) = 11.30$, $p < .001$, suggesting that participants desired some degree of wealth inequality.

### Comparing past, present, and future inequality

A repeated measures ANOVA with Greenhouse-Geisser correction showed reliable differences between the three esti-mates, $F(2, 453) = 10.01$, $p < .001$, $\eta^2_p = .04$. Again, the linear trend was significant, $F(1, 232) = 21.14$, $p < .001$, $\eta^2_p = .08$,

in line with the hypothesized decline model. Pairwise mean comparisons revealed that participants estimated present inequality ($M = 49.14\%$, $SD = 15.81\%$) as significantly lower than future wealth inequality ($M = 52.35\%$, $SD = 18.07\%$), $t(232) = -2.88$, $p = .005$, $d = -.34$. Present inequality, however, was not estimated as significantly higher than past wealth inequality ($M = 47.83\%$, $SD = 15.18\%$), $t(232) = 1.27$, $p = .203$, $d = .09$.

### Comparing 2016 and 2021 estimates across studies

Participants' estimates for future wealth inequality in 2026 in Study 1 ($M = 51.60\%$, $SD = 18.87\%$) and Study 2 ($M = 52.35\%$, $SD = 18.07\%$) did not differ, $t(554) = -.47$, $p = .636$. Similarly, estimates for 2016 in Study 1 (i.e., present wealth inequality; $M = 46.47\%$, $SD = 17.37\%$) and Study 2 (i.e., past wealth inequality; $M = 47.83\%$, $SD = 15.18\%$) showed again no differences, $t(554) = -.97$, $p = .355$. Thus, estimates of the two cohorts were very similar.

### Discussion

Study 2 replicated results of Study 1 on people's generalized underestimation of wealth inequality (e.g., [7,27]). Although we were not able to compare participants' estimates to actual data, again we found that participants preferred a society with less wealth inequality, but without achieving full equality.

In line with the predicted decline model, wealth inequality was seen to increase in a linear fashion over time, although in the present study the difference between past and present was not significant. This may be due to the fact that the time lag was reduced from 10 to 5 years; it would be quite logical if changes in the distribution of wealth were perceived as stronger the longer the time period under consideration. Similarly, no differences emerged when comparing our 2016 and our 2021 data. We will further address these results in the general discussion.

### General discussion

Research on wealth inequality perception has highlighted that laypeople generally misperceive the magnitude of wealth inequality. These studies, however, never attempted to extend these findings to perceptions of wealth inequality over time. Taking inspiration from the literature on collective memory bias [10], we aimed at filling this gap by assessing perceptions of past, present, and future wealth distributions, and testing the existence of a decline model (i.e., optimism towards past wealth inequality, pessimism towards future wealth inequality).

Across two studies, results showed that participants greatly underestimated present levels of wealth inequality and expressed a preference for an ideal society with smaller (but not zero) levels of wealth inequality. These findings extend previous evidence on wealth inequality perception (e.g., [7,29]) to the Italian context, showing that people may be inaccurate in their estimates of wealth inequality, but have clear ideas about what their ideal society should look like.

More central to the aims of our research are results that confirm our collective memory bias hypothesis. Participants tended to (correctly) perceive an increase in wealth inequality over time. They thought that society had been more equal in the past than it was in the present (although this comparison was statistically significant only in our 2016 sample) and they also believed that it would become even more unequal in the future. Interestingly, estimates provided by participants in Study 1 and Study 2 for matching time periods (i.e., 2016 and 2026) did not differ between studies. Thus, participants in Study 1 and Study 2 produced, on average, very similar estimates for the same time periods, except for the fact that differences between past and present estimates were larger in our 2016 compared to our 2021 sample.

Two explanations may be offered for this finding. Historically, the international financial crisis of 2007, which gave rise to the Great Recession of 2008, was located between the "past" (2006) and the "present" (2016) point in time in Study 1, which may have polarized estimates for the two time-stamps. In contrast, the two time-stamps in Study 2 (2016 as past and 2021 as present) both fell into the same post-financial crisis period, possibly making them look more similar. A more parsimonious explanation is that the time difference between "past" and "present" (and between "present" and "future")

was 10 years in Study 1, but only 5 years in Study 2. Assuming a linear (mental) timeline, mean differences between two points in time will logically be larger as the temporal distance increases.

Considering the above findings together, one may advance the hypothesis that participants rely on a rational mechanism in which they project linearly how wealth inequality grows (or declines) over time, reflecting a decline or an improvement model, and that they plot their estimates for specific points in time on this linear projection. Such mental schemas are likely to prompt relative (rather than absolute) estimates of wealth gaps – which were indeed found to be more precise than absolute ones [34], even for topics related to the economy (e.g., social mobility; [35]).

Additional proof for this reasoning comes from the European ranking task that was not discussed in the main text but is available in the Supporting Information. Here, participants evaluated inequality levels by comparing different countries (rather than simply estimating Italian wealth gaps alone), showing relatively accurate rankings. Participants were relatively accurate in ordering countries with respect to wealth inequality, placing 6 out of 9 countries in the correct order in Study 1, and 5 out of 9 countries in the correct order in Study 2. They also placed Italy close to its actual rank (roughly 7th place rather than 6th place for Study 1, and between 6th and 7th place rather than 8th place in Study 2). This may reflect the fact that ranking tasks are easier than estimates of absolute values (e.g., [47]), and that people are able to grasp the relative standing of different nations even when they have only a very limited understanding of absolute levels of inequality.

Together, our results confirm the idea that participants are highly inaccurate in guessing absolute levels of wealth inequality, whereas their relative judgments are surprisingly accurate, as visible in Fig 2. Estimates remained consistently below actual wealth gaps (showing a systematic underestimation) but the trajectories of actual and estimated wealth inequality over time showed almost identical slopes.

Overall, these findings suggest that public communication and educational campaigns about wealth inequality should address people's underlying perceptions and cognitive biases. Because individuals tend to underestimate current inequality, idealize the past, and expect a worsening future, communication strategies should aim to correct these misperceptions by presenting clear, relatable evidence of actual inequality trends. At the same time, messages should counteract pessimism by emphasizing that inequality is not inevitable and can be reduced through collective action and policy choices. Since people seem to understand relative differences better than absolute figures, comparative presentations—showing how inequality varies across countries or over time—may be particularly effective. Therefore, these results highlight the importance of designing campaigns that combine accurate data, emotional resonance, and a sense of agency, encouraging citizens to see inequality as both a real and changeable social issue.

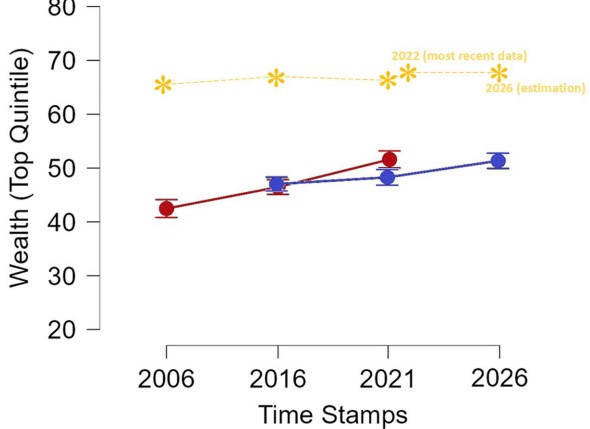

**Fig 2. Comparison of Actual and Estimated Wealth Inequality Across Time Stamps (Study 1 and 2).** Note: Red=Average Study 1, Blue=Average Study 2, Yellow=Actual Inequality in Italy.

## Limits and future directions

Being the first of its kind, our research naturally has some limitations. Firstly, Study 1 relied on convenience sampling via social media, which may have introduced self-selection bias and constrained the generalizability of the findings. Individuals who choose to participate in online studies are often younger, more educated, and more digitally active than the general population, potentially limiting the representativeness of the sample. Although Study 2 recruited participants through Prolific Academic, which provides greater demographic diversity and experimental control, this sample may still not accurately reflect the broader Italian population. As a result, the external validity of our results should be interpreted with caution. Future research could enhance the generalizability of our results by adopting more representative sampling approaches, such as using sample weights. Employing stratified or quota sampling procedures, or recruiting from nationally representative panels, would allow for broader demographic coverage and more robust population inferences. Additionally, combining online recruitment with offline data collection (e.g., through community organizations) could help reach underrepresented groups, thereby improving the inclusiveness and ecological validity of results.

On a methodological note, one could argue that the trend of underestimation found in our studies may be an artifact of the scale anchoring, as the correct option was placed close to the end-point of the scale: thus the design may have driven people to provide underestimates, for example if people assumed the correct response was likely to be in the middle of the scale. Thus, future replications of this design should adopt alternative measures of perceived inequality (e.g., [48]). Moreover, the pie chart task used to assess perceptions of wealth inequality constrained participants' responses by not allowing the selection of inequality levels exceeding the actual distribution. This design feature may have restricted participants' ability to express more pessimistic views about levels of wealth inequality in Italy. As a result, the measure may underestimate perceived inequality among individuals who view societal disparities as more extreme than they are in reality. In addition, participants who selected the "even higher" option when reporting past or future inequality were excluded from the analyses. Excluding these participants ($n = 31$ in Study 1 and $n = 13$ in Study 2) may have led to a loss of potentially informative data from individuals perceiving exceptionally high wealth inequality in Italy. Future research should address these limitations by employing more flexible measures of perceived inequality, such as, for instance, tasks allowing participants to freely adjust levels of wealth inequality beyond the actual distribution. Additionally, incorporating complementary qualitative or narrative methods could provide deeper insight into the subjective meanings and emotional underpinnings of perceived levels of wealth inequality. Together, such approaches would yield a more nuanced and comprehensive understanding of how individuals conceptualize and evaluate social disparities. Future research could also conduct sensitivity analyses or adopt alternative coding strategies—such as assigning a ceiling value to "even higher responses"—to better capture the upper bounds of perceived inequality. This would allow for a more comprehensive assessment of variability in perceptions and ensure that the full range of respondents' views is represented.

A further limitation concerns the change in temporal framing between studies. Study 1 assessed perceptions of inequality using 10-year intervals, whereas Study 2 employed 5-year intervals. This discrepancy introduces a methodological inconsistency that may affect the comparability of results across studies. Shorter intervals in Study 2 may have influenced both the magnitude and temporal focus of perceived levels of inequality trends. Conversely, the broader 10-year intervals used in Study 1 may have encouraged more general or retrospective evaluations, capturing longer-term perceptions of societal change. Future research should aim to standardize temporal framing across studies to ensure comparability and interpretative consistency. Alternatively, researchers could systematically manipulate interval length within a single study to examine how temporal framing influences perceptions of wealth inequality and social change. Such designs would help clarify whether observed differences reflect genuine variation in perceptions or are partly attributable to methodological differences in measurement.

Finally, both studies were correlational by design; future replications may attempt to assess directionality by manipulating the (mental) decline vs. improvement model and investigate their impact on the estimated past, present, and future wealth inequalities. Future studies could also explore additional moderating or mediating variables. For example,

Yamashiro and Roediger III [10] investigated how participants' convictions about their country (i.e., a measure of American exceptionalism assessing beliefs of the US as a unique and superior nation) may shape collective memory biases, whereas Hirst and colleagues [49,50] tested the role played by national identity. This addition may provide further proof for our proposed decline model, given that Italians are particularly pessimistic about national economy trends [51]. Moreover, while the assessment of wealth estimations in the Italian context represents a novelty – particularly when compared to previous literature (which is mainly US-centric) – future studies should try to replicate our findings on temporal memory biases in wealth inequality perceptions across a wider range of countries (including non-Western countries).

## Conclusion

Most psychological literature on economic inequality has focused on misperception of present wealth and income gaps. In line with this literature, we found that people tend to greatly underestimate current levels of wealth inequality. However, although people's perceptions were inaccurate in absolute terms (i.e., they greatly underestimate levels of past, present, and future inequality when compared to actual data), they correctly predicted wealth inequality trends over time, following a decline model of wealth inequality that was supported by actual data (e.g., [1]). Designers of future information campaigns may well be advised to focus on time trends, as this would match people's mental representation of inequality – which could be crucial in predicting, for example, their attitudes towards policies aimed at reducing inequality itself.

## Supporting information

**S1 File. Supporting information.** This file contains all Supporting Information figures, tables, and additional analyses not included in the manuscript.
(DOCX)

## Author contributions

**Conceptualization:** Andrea Scatolon, Silvia Galdi, Carmen Cervone, Anne Maass.

**Data curation:** Andrea Scatolon.

**Formal analysis:** Andrea Scatolon, Silvia Galdi, Carmen Cervone, Anne Maass.

**Funding acquisition:** Anne Maass.

**Investigation:** Andrea Scatolon, Silvia Galdi, Carmen Cervone, Lucia Mannetti, Anne Maass.

**Methodology:** Silvia Galdi, Anne Maass.

**Project administration:** Andrea Scatolon, Silvia Galdi, Anne Maass.

**Supervision:** Silvia Galdi, Anne Maass.

**Visualization:** Andrea Scatolon.

**Writing – original draft:** Andrea Scatolon.

**Writing – review & editing:** Andrea Scatolon, Silvia Galdi, Carmen Cervone, Anne Maass.

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
