## [Decision Letter · Decision Letter 0]

25 Sep 2025

Dear Dr. Scatolon,

Thank you for submitting your manuscript to PLOS ONE. After careful consideration, we feel that it has merit but does not fully meet PLOS ONE’s publication criteria as it currently stands. Therefore, we invite you to submit a revised version of the manuscript that addresses the points raised during the review process.

I have now received the comments of two reviewers. After carefully reading them, I believe that the paper could be considered for publication after additional revisions.

In addition to the reviewers’ suggestions, I would encourage you to place greater emphasis on the potential biases of using a convenience sampling. In particular, please address the risk of selection bias due to non-random non-response, as well as the limitations arising from the absence of sample weights in your data.

We look forward to receiving your revised manuscript.

Kind regards,

Pablo Gutierrez Cubillos

Academic Editor

PLOS ONE

Additional Editor Comments:

I have now received the comments of two reviewers. After carefully reading them, I believe that the paper could be considered for publication after additional revisions.

In addition to the reviewers’ suggestions, I would encourage you to place greater emphasis on the potential biases of using a convenience sampling. In particular, please address the risk of selection bias due to non-random non-response, as well as the limitations arising from the absence of sample weights in your data.

Reviewers' comments:

Reviewer's Responses to Questions

**Comments to the Author**

1. Is the manuscript technically sound, and do the data support the conclusions?

Reviewer #1: Yes

Reviewer #2: Yes

2. Has the statistical analysis been performed appropriately and rigorously?

Reviewer #1: Yes

Reviewer #2: Yes

3. Have the authors made all data underlying the findings in their manuscript fully available?

Reviewer #1: Yes

Reviewer #2: Yes

4. Is the manuscript presented in an intelligible fashion and written in standard English?

Reviewer #1: Yes

Reviewer #2: Yes

Reviewer #1: This is an overall convincing article. The methodology is rigorous, and the findings are presented clearly and concisely. The authors have made a full reflection on limitations, but this part can be expanded further.

Reviewer #2: Your manuscript describes an interesting and insightful piece of research. The data support your conclusions well. Participants consistently underestimated current wealth inequality, and their perceptions followed a linear trajectory over time, consistent with actual trends. The interpretation of this pattern as a “decline model” is well-reasoned and supported by both empirical data and theoretical framing.

However, addressing the following points could strengthen the manuscript:

• Sampling Bias: Study 1 relied on convenience sampling via social media, which may limit generalisability. While Study 2 used Prolific Academic, a more controlled platform, it still may not fully represent the Italian population. Please discuss this limitation more explicitly in your manuscript. Also in the study limitation section, could you suggest how future studies can address sampling methods limitations to improve generalisability, e.g., use of nationally representative samples?

• Measurement Constraints: The pie chart task used to assess perceptions of inequality is intuitive and visually engaging. However, the response options did not allow participants to express perceptions of inequality greater than the actual level. This may have truncated responses and limited the full expression of pessimism. Consider discussing this limitation more critically and suggesting alternative approaches for future research.

• Temporal Framing: The change from 10-year intervals in Study 1 to 5-year intervals in Study 2 introduces a methodological inconsistency. While you acknowledge this in the discussion, a more detailed explanation of how this might affect comparability would be beneficial.

• Handling of “Even Higher” Responses: Participants who selected the “even higher” option for past and future inequality were excluded from analysis. While this is understandable due to quantification challenges, a sensitivity analysis or alternative coding strategy (e.g., assigning a ceiling value) could provide additional insight into the extent of perceived inequality. This is only a suggestion for future papers.

• Exclusion Criteria: Provide a more detailed rationale for excluding participants who selected the “even higher” option (n=13), and consider alternative ways to include their responses.

• Terminology: The term “decline model” is central to your manuscript, but could be more clearly defined early on. Clarifying that this refers to increasing inequality as a negative societal trend would help readers unfamiliar with the concept.

• Policy Implications: Expand your discussion on how these findings could inform public communication strategies or educational campaigns aimed at improving awareness of inequality.

**Do you want your identity to be public for this peer review?** For information about this choice, including consent withdrawal, please see our Privacy Policy

Reviewer #1: No

Reviewer #2: No

---

## [Author Response · Author response to Decision Letter 1]

23 Nov 2025

Dear Dr. Pablo Gutierrez Cubillos,

Please find enclosed for your consideration the revised version of the manuscript entitled “Beyond the perception of present economic inequality: How people construe past, present and future wealth gaps”, which we would like to re-submit for publication to PLOS ONE (MS number: PONE-D-25-31479).

We have now revised the manuscript along with the Editor’s comments, as well as Reviewer #1’s and Reviewer #2’s comments. A complete list of the changes can be found in the following pages, with changes to the manuscript in red, both in this document and in the revised manuscript.

We believe that the manuscript has improved as a result of this review process, and we thank both you and the reviewers for your contributions. We hope you will find the current version of the manuscript worthy of publication in PLOS ONE. We thank you once again for your attention and look forward to hearing back from you at your earliest convenience.

Yours sincerely,

Andrea Scatolon

###

1. Please ensure that your manuscript meets PLOS ONE's style requirements, including those for file naming. The PLOS ONE style templates can be found at Https://journals.plos.org/plosone/s/file?id=wjVg/PLOSOne_formatting_sample_main_body.pdf and https://journals.plos.org/plosone/s/file?id=ba62/PLOSOne_formatting_sample_title_authors_affiliations.pdf

The ethics statement was previously included in the Present Research section. It was moved to the Methods section of each study accordingly (page 10, line 209; page 17, line 379).

Neither Reviewer #1 nor Reviewer #2 recommended any specific citation; therefore, no changes were made.

The reference list was checked, and we confirm that it is complete and follows PLOS ONE’s guidelines. No changes to the reference list were made.

Additional Editor Comments:

In addition to the reviewers’ suggestions, I would encourage you to place greater emphasis on the potential biases of using a convenience sampling. In particular, please address the risk of selection bias due to non-random non-response, as well as the limitations arising from the absence of sample weights in your data.

We thank the Editor for this insight. We addressed this point as follows (page 24, line 525):

Being the first of its kind, our research naturally has some limitations. Firstly, Study 1 relied on convenience sampling via social media, which may have introduced self-selection bias and constrained the generalizability of the findings. Individuals who choose to participate in online studies are often younger, more educated, and more digitally active than the general population, potentially limiting the representativeness of the sample. Although Study 2 recruited participants through Prolific Academic, which provides greater demographic diversity and experimental control, this sample may still not accurately reflect the broader Italian population. As a result, the external validity of our results should be interpreted with caution. Future research could enhance the generalizability of our results by adopting more representative sampling approaches, such as using sample weights. Employing stratified or quota sampling procedures, or recruiting from nationally representative panels, would allow for broader demographic coverage and more robust population inferences. Additionally, combining online recruitment with offline data collection (e.g., through community organizations) could help reach underrepresented groups, thereby improving the inclusiveness and ecological validity of results.

Reviewer #2:

Your manuscript describes an interesting and insightful piece of research. The data supports your conclusions well. Participants consistently underestimated current wealth inequality, and their perceptions followed a linear trajectory over time, consistent with actual trends. The interpretation of this pattern as a “decline model” is well-reasoned and supported by both empirical data and theoretical framing. However, addressing the following points could strengthen the manuscript:

1) Sampling Bias: Study 1 relied on convenience sampling via social media, which may limit generalisability. While Study 2 used Prolific Academic, a more controlled platform, it still may not fully represent the Italian population. Please discuss this limitation more explicitly in your manuscript. Also in the study limitation section, could you suggest how future studies can address sampling methods limitations to improve generalisability, e.g., use of nationally representative samples?

We thank Reviewer #2 for pointing this out. We addressed this at the beginning of the Limits and Future Directions section (page 24, line 525):

Being the first of its kind, our research naturally has some limitations. Firstly, Study 1 relied on convenience sampling via social media, which may have introduced self-selection bias and constrained the generalizability of the findings. Individuals who choose to participate in online studies are often younger, more educated, and more digitally active than the general population, potentially limiting the representativeness of the sample. Although Study 2 recruited participants through Prolific Academic, which provides greater demographic diversity and experimental control, this sample may still not accurately reflect the broader Italian population. As a result, the external validity of our results should be interpreted with caution. Future research could enhance the generalizability of our results by adopting more representative sampling approaches, such as using sample weights. Employing stratified or quota sampling procedures, or recruiting from nationally representative panels, would allow for broader demographic coverage and more robust population inferences. Additionally, combining online recruitment with offline data collection (e.g., through community organizations) could help reach underrepresented groups, thereby improving the inclusiveness and ecological validity of results.

2) Measurement Constraints: The pie chart task used to assess perceptions of inequality is intuitive and visually engaging. However, the response options did not allow participants to express perceptions of inequality greater than the actual level. This may have truncated responses and limited the full expression of pessimism. Consider discussing this limitation more critically and suggesting alternative approaches for future research.

We thank Reviewer #2 for these considerations. We addressed this limitation in the Limits and Future Directions section (page 25, line 550):

Moreover, the pie chart task used to assess perceptions of wealth inequality constrained participants’ responses by not allowing the selection of inequality levels exceeding the actual distribution. This design feature may have restricted participants’ ability to express more pessimistic views about levels of wealth inequality in Italy. As a result, the measure may underestimate perceived inequality among individuals who view societal disparities as more extreme than they are in reality.

3) Temporal Framing: The change from 10-year intervals in Study 1 to 5-year intervals in Study 2 introduces a methodological inconsistency. While you acknowledge this in the discussion, a more detailed explanation of how this might affect comparability would be beneficial.

We further addressed this issue in the Limits and Future Direction section (page 26, line 570)

A further limitation concerns the change in temporal framing between studies. Study 1 assessed perceptions of inequality using 10-year intervals, whereas Study 2 employed 5-year intervals. This discrepancy introduces a methodological inconsistency that may affect the comparability of results across studies. Shorter intervals in Study 2 may have influenced both the magnitude and temporal focus of perceived levels of inequality trends. Conversely, the broader 10-year intervals used in Study 1 may have encouraged more general or retrospective evaluations, capturing longer-term perceptions of societal change. Future research should aim to standardize temporal framing across studies to ensure comparability and interpretative consistency. Alternatively, researchers could systematically manipulate interval length within a single study to examine how temporal framing influences perceptions of wealth inequality and social change. Such designs would help clarify whether observed differences reflect genuine variation in perceptions or are partly attributable to methodological differences in measurement.

4 and 5) Handling of “Even Higher” Responses: Participants who selected the “even higher” option for past and future inequality were excluded from analysis. While this is understandable due to quantification challenges, a sensitivity analysis or alternative coding strategy (e.g., assigning a ceiling value) could provide additional insight into the extent of perceived inequality. This is only a suggestion for future papers. Exclusion Criteria: Provide a more detailed rationale for excluding participants who selected the “even higher” option (n=13), and consider alternative ways to include their responses.

We thank Reviewer #2 for this insight, which we took into consideration in our Limits and Future Direction section (page 25, line 555):

In addition, participants who selected the “even higher” option when reporting past or future inequality were excluded from the analyses. Excluding these participants (n = 31 in Study 1 and n = 13 in Study 2) may have led to a loss of potentially informative data from individuals perceiving exceptionally high wealth inequality in Italy. Future research should address these limitations by employing more flexible measures of perceived inequality, such as, for instance, tasks allowing participants to freely adjust levels of wealth inequality beyond the actual distribution. Additionally, incorporating complementary qualitative or narrative methods could provide deeper insight into the subjective meanings and emotional underpinnings of perceived levels of wealth inequality. Together, such approaches would yield a more nuanced and comprehensive understanding of how individuals conceptualize and evaluate social disparities. Future research could also conduct sensitivity analyses or adopt alternative coding strategies—such as assigning a ceiling value to “even higher responses”—to better capture the upper bounds of perceived inequality. This would allow for a more comprehensive assessment of variability in perceptions and ensure that the full range of respondents’ views is represented.

6) Terminology: The term “decline model” is central to your manuscript, but could be more clearly defined early on. Clarifying that this refers to increasing inequality as a negative societal trend would help readers unfamiliar with the concept.

To address this point, we anticipated what we consider as “decline model” (page 7, line 157):

Based on previous literature, we have reasons to believe that such a connection exists: for example, consider how previous literature (11) has provided evidence for a decline model (which we intend as referring to an increase in inequality as a negative societal trend) when assessing memory biases, and how a specific variable, namely perceived societal anomie (i.e., perceived deregulation and disintegration of one’s society), worked as a buffer for such biases.

7) Policy Implications: Expand your discussion on how these findings could inform public communication strategies or educational campaigns aimed at improving awareness of inequality.

We addressed this issue in the General Discussion section (page 23, line 512):

Overall, these findings suggest that public communication and educational campaigns about wealth inequality should address people’s underlying perceptions and cognitive biases. Because individuals tend to underestimate current inequality, idealize the past, and expect a worsening future, communication strategies should aim to correct these misperceptions by presenting clear, relatable evidence of actual inequality trends. At the same time, messages should counteract pessimism by emphasizing that inequality is not inevitable and can be reduced through collective action and policy choices. Since people seem to understand relative differences better than absolute figures, comparative presentations—showing how inequality varies across countries or over time—may be particularly effective. Therefore, these results highlight the importance of designing campaigns that combine accurate data, emotional resonance, and a sense of agency, encouraging citizens to see inequality as both a real and changeable social issue.

We hope that the present revised manuscript is now suitable for publication in PLOS ONE, and we look forward to hearing your reactions.

Best wishes

---

## [Editor Report · Decision Letter 1]

18 Dec 2025

Beyond the perception of present economic inequality: How people construe past, present and future wealth gaps

PONE-D-25-31479R1

Dear Dr. Scatolon,

We’re pleased to inform you that your manuscript has been judged scientifically suitable for publication and will be formally accepted for publication once it meets all outstanding technical requirements.

Kind regards,

Pablo Gutierrez Cubillos

Academic Editor

PLOS One
---

## [Editor Report · Acceptance letter]

PONE-D-25-31479R1

PLOS One

Dear Dr. Scatolon,

I'm pleased to inform you that your manuscript has been deemed suitable for publication in PLOS One. Congratulations! Your manuscript is now being handed over to our production team.

Kind regards,

on behalf of

Dr. Pablo Gutierrez Cubillos

Academic Editor

PLOS One